# Biomimetic Design of Turbine Blades for Ocean Current Power Generation

**DOI:** 10.3390/biomimetics8010118

**Published:** 2023-03-11

**Authors:** Enrique Eduardo Hernández Montoya, Edgar Mendoza, Eize J. Stamhuis

**Affiliations:** 1Biomimetic Group, Energy and Sustainability Research Institute Groningen, Faculty of Science and Engineering, University of Groningen, 9712 CP Groningen, The Netherlands; 2Instituto de Ingeniería, Grupo de Ingeniería de Costas y Puertos, Facultad de Ingeniería, Universidad Nacional Autónoma de México, Ciudad Universitaria 04510, Ciudad de México, Mexico

**Keywords:** biomimetics, bio-inspired blades, marine turbines model, aerodynamic properties, hydrodynamic properties, wind tunnel measurements, flow tank facility, Betz limit

## Abstract

The enhancement of energy technology and innovation play a crucial role in order to meet the challenges related to global warming in the coming decades. Inspired by bird wings, the performance of a bio-inspired blade assembled to a marine turbine model, is examined. Following a biomimetic pathway, the aerodynamic performance of the bird wings of the species Common Guillemot (*Uria aalge*) was tested in a wind tunnel laboratory. Based on our results, we derived a bio-inspired blade model by following a laser scanning method. Lastly, the bio-inspired blades were assembled to a marine turbine model and tested in a large flow tank facility. We found efficiencies ($C_{p}$) up to 0.3 which is around 53% of the maximum power that can be expected from the turbine model according to the Betz approach. Our findings are analyzed in the discussion section as well as considerations for future research.

## 1. Introduction

In recent years, the consequences of global warming have been clearly documented all over the world. Global warming is mostly caused by overpopulation and the increasing global demand toward energy. As a way to decrease the probability of catastrophic scenarios related to climate changes, i.e., tsunamis, earthquakes and floods, the scientific community, governments and industrial sectors have allocated resources and strengthen global agreements in order to upgrade renewable energy. The International Energy Agency (IEA) reported in 2020 what is needed to accomplish net zero emissions by 2050, specifying that a sustained and complete shift to clean energy technologies is needed [1]. These technologies should provide the energy services we need and should not emit green-house gases or pollute air, land or water. Therefore, the enhancement of energy technology and innovation might be crucial in the coming decade [1]. Renewable energies such as wind power and solar photovoltaics (solar PV) are an example of success as clean energy technologies. Nowadays, wind power and solar PV account for 17% and 3.6%, respectively, of global electricity needs [2,3]. However, such technologies started to develop several decades ago at the investment of significant public funding. The low pace at which new technologies are evolving shows the importance to develop more efficient devices as soon as possible [1].

A thriving source that has great potential for generation of electric power is the Marine Renewable Energies (MRE). MRE uses hydrokinetic energy that is generated in the oceans by e.g., tides, ocean currents, waves and thermal and salinity gradients. Harnessing this type of energy is projected to develop noticeably in the coming decades since half of the global population living in cities with over 100,000 inhabitants are within 100 km from the coast and could therefore potentially benefit from MRE [4]. Principally in western Europe, research and development of hydrokinetic energy has been directed to tidal and wave technology due to its abundance around northern latitudes. Ocean currents could provide large advantages when compared to tidal currents as they show nearly continuous and unidirectional flow. Nevertheless, ocean current device developers have faced difficulties designing economical mooring systems and maintenance procedures for deep water sites [5,6]. However, recent studies show the possibility of harnessing open ocean currents around coastal areas [7].

Ocean currents becomes a potential promising alternative when the high current area is associated with geomorphological and seabed topography features such as straits and channels. The Cozumel Channel, located in the Mexican Caribbean, forms such a passageway, transporting circa 5 Sv (5 Mm^3^ s^−1^), entailing about 20% of the Yucatan Current flows northward parallel to the Yucatán Península. One specific area, located outside the delimitation framed by protected marine areas, navigational channels and tourism activities shows a potential zone which extends to approximately 70 ha of seabed suitable for the deployment of marine energy devices [5]. A recent assessment done in the area of interest measured average stream velocities of 0.93 m s^−1^ (σ=±0.23 m s^−1^) and peak velocities of ≈1.2−1.6 m s^−1^. Available technology is currently not sufficient to be immediately installed considering the flow being a relatively slow-moving oceanic current [7]. Horizontal axis tidal turbines (HATTs) have been identified as the most feasible device since its theoretical background is based on the successful results of wind turbine technologies which also contributes to minimize economical costs and learning procedures [5]. However, in order to optimize the costs of energy generation, further studies are required to develop an innovative and efficient turbine design for these particular current characteristics.

To develop highly efficient small wind turbines (SWTs), several studies have been conducted applying biomimetic methods, that is the case of bio-inspired designs and optimization processes. For example, Ikeda et al., found that a bio-inspired flex blade outperforms a conventional blade by 8.1%, based on blade element momentum theory [8]. Tian et al., 2017 found similar results when developing a bio-inspired turbine blade showing an efficiency increase of 12% in wind turbine tests when compared to a standard blade [9]. Moreover, experimental methodology comparing different bird species properties has been developed in order to identify improvements in efficiency when compared to conventional SWTs blades [10,11,12,13]. These studies show a considerable increase of efficiency in power output of bio-inspired blades compared to conventional blades. In the present study, the increase in efficiency of a biomimetic blade design for marine turbines is examined using a biomimetic experimental approach. For this, we analyze the aerodynamic properties of a carefully preserved bird wing of a suitable species. Based on our findings we use a laser scan method to obtain the wing profiles taken under aerodynamic load at several positions along the wing span during experimental tests. Lastly, we derive a full 3D turbine blade which is prototyped and subsequently installed on a marine turbine model and tested in a flow tank facility at a range of flow velocities. Results are graphed in terms of power (P), flow speed (u), power coefficient ($C_{p}$) and tip speed ratio (TSR) and compared to theoretical values.

## 2. Materials and Methods

### 2.1. Preparation of the Wing

The first step of the biomimetic method was the selection of a suitable bird species of which the wing aero/hydro-dynamics have a certain similarity with a marine turbine blade in function. The species selected was the Common Guillemot (*Uria aalge*). In selecting this species two criteria were mainly taken into consideration: The first one is the high wing loading of around 2 gr/cm^2^. Wing loading was taken as the total mass of the bird divided by its total wing area. In air, the Guillemot is not very agile and take-off is difficult for them. However, they have sacrificed flight efficiency in favour of diving during their evolutionary pathway. The Common Guillemot is a pursuit-diver that forages for food by swimming underwater using its wings for propulsion [14] which makes it very suitable for our study. The second criterium was the availability of a carcass of a suitable bird species. In the present study no bird was sacrificed but a fresh carcass of a beached individual was used. The availability of potentially suitable bird species was regularly tracked based on the frozen stock availability of the nearby animal shelter until a suitable specimen was available.

Once the bird is chosen, the wing must be separated from the body. The wing is connected to the main body through the shoulder joint which is particularly important to preserve in order to position the wing in the required final shape. To identify the shoulder joint, it is needed to follow the wing bones from tip to the base, identify the shoulder joint and subsequently cut the wing loose by cutting through the shoulder joint itself. The wing can now be set in a glide position under aerodynamic load and then mounted in that position to a tailor-made PVC plate using tie-raps. This specific condition results into the unique shape of interest of the wing to be studied under normal aerodynamic load. The process described above is shown in Figure 1.

The wing in the defined load shape was preserved following a freeze-drying method utilized to preserve specimens for teaching collections (see Shoffner in 2013) [15]. Our freeze dryer consists of a refrigerated specimen chamber, a refrigerated condenser and a vacuum pump. The specimens are kept at below zero degrees (°C) temperatures within the closed-off chamber and the vacuum pump lowers the pressure within the chamber to facilitate sublimation and the removal of water as vapor from the chamber. Shoffner indicates quantitative and qualitative parameters to evaluate a successful freeze-drying specimen: The quantitative parameter is a mass loss of at least 32.5%. This parameter was defined after the freeze-drying method was applied to 5 bird wings where all of them showed the percentage of mass loss aforementioned. The qualitative consideration is that the texture of the specimen must be seen as dry, thin and brittle [15]. For the present study a benchtop freeze-dryer was used (Labconco FreeZone^®^, Kansas City, MO, USA). This equipment operates at a temperature of approximately −54 °C and a pressure of approximately 0.030 mbar. Different sizes of bird wings were processed in the freeze-drying machine and in order to guarantee the success of the process they were kept in there for at least 12 days, see Figure 1c. The measured values before and after being freeze-dried are shown in Table 1. Once the wings were taken out, the quantitative and qualitative parameters were calculated and examined respectively. As both parameters were accomplished, the freeze-drying process was considered as successful.

Once the wings are successfully dried, they are ready to be attached to a plastic base for the measurement set-up in the wind tunnel. Considering the size of the humerus of the bird, a corresponding diameter of the plastic tube is chosen and glued to the ball-end shape of the bone using hard polyurethane foam, which is known for its compatibility with water-based materials and form stability. On the other side of the plastic tube a stainless-steel tube is attached with epoxy glue taking good care of a straight positioning. Lastly, a round transparent acrylic plate was glued to the plastic tube mimicking the body of the bird in order to replicate the aerodynamic behavior while flying, see Figure 1d.

### 2.2. Wind tunnel Measurements

To study the aerodynamic properties of the bird wings, wind tunnel laboratory tests were developed. In 2015 the Energy & Sustainability Research Institute Groningen (ESRIG —FSC—University of Groningen) re-installed an obsolete wood-based high-quality tunnel originally part of the Department of Aerodynamics of the Hochschule, Bremen. Housed in the Nuclear Physics Building—Campus Groningen (KVI), the wind tunnel has been modified from its previous version. Nowadays, the facility has three measurement sections and operates as a closed-loop tunnel. The large section is mostly used for low-speed testing of experimental small wind turbines, the middle-speed section is used for bird wings and truck models, and the small section one for high-speed blade profile testing. The maximum flow speeds of each section are (ca): 5, 15 and 45 m s^−1^, i.e., 18, 54 and 162 km h^−1^ respectively [16]. Dimensions of each section and a side view of the wind tunnel can be seen in Figure 2.

In accordance with the load position, size of the bird wing and maximum flow speed, the second section of the wind tunnel was selected to perform the force measurements. The stainless-steel tube from the bird wing fixation was attached via an aluminum coupler to a stepper motor (Stepperonline^®^, NY, USA, Nema 11 Bipolar L = 51 mm w/gear ratio 27:1, Planetary Gearbox) which in turn was mounted to the force sensor (RFS^®^, Radial dual-axis force sensor 150 XY–10N, accuracy class 0.25). Both components were covered by a 3D printed streamliner with the objective of avoiding disturbances of the air flow. The streamliner was attached to a thick metal plate coming out of a slit in the polycarbonate tunnel bottom. All the wiring (stepper motor and force sensor) was fed through the slit to measurement and control equipment. The full set-up can be seen in Figure 3.

The force sensor was calibrated before and after every measurement session by adding weights to the stainless-steel tube in X and Y directions corresponding to drag and lift forces, respectively. Calibrations for weights of 0, 1, 2, 5, 10, 20, 50, 100 and 200 g were measured to confirm the linearity of the force sensor and this was accepted when a correlation $\left(R^{2}\right)$ of at least 0.99 was achieved. Wind speeds of 4, 6, 8, 10 and 12 m s^−1^ were set in the wind tunnel at each angle of attack from −10° to +40° with steps of 2°. The force pickup and processing system (Honigmann Tensiotron TS621 amplifiers, Honigmann USB 50 A/D 16-bit Data Acquisition System, Honigmann HCC Easy registration and analysis software. Specifications and uncertainty values can be consulted in [17]) was set to measure up to 1000 samples per second for a period of 10 s for every angle of attack. In order to produce reliable data every measurement set (i.e., each combination of wind speed and angle of attack) was measured 5 times for left as well as for the right wing of the bird.

In order to calculate the real lift and drag forces, the values taken from the sensor measurements were transformed to weight values using the linear interpolation graph used for calibration. Once in grams units, the equivalence of 1 g = 0.00981 N was utilized to calculate the measured lift and drag forces. Prior to the wing measurements, the same set-up using an exact same model but without the wing was tested for each wind speed in order to calculate the additional lift and drag forces generated only by the adapters and measurement tools. By subtracting these values from the measured lift and drag values the real lift and drag forces were obtained. To be able to compare the characteristics of our measurements to literature and to standard airfoils, all lift and drag values were recalculated to nondimensional lift- and drag- coefficients, using Equations (1) and (2), respectively [18].
(1)$$C_{L}=\frac{2F_{L}}{\rho \;v^{2}\;A}$$
(2)$$C_{D}=\frac{2F_{D}}{\rho \;v^{2}\;A}$$
where $C_{L}$ and $C_{D}$ stands for lift and drag coefficients, $F_{L}$ and $F_{D}$ for lift and drag forces [N], ρ for air density [kg m^−3^], v for velocity [m s^−1^] and A for the wing planform area [m^2^].

### 2.3. Deriving a Turbine Blade from Bird Wing Profiles

Once the higher angle of attack and flow velocity combination was identified, a laser technique was chosen to obtain the bird wing profiles. A similar laser scan methodology as used by Lentink et al. [19], was followed. The main objective of this procedure was to obtain profiles from the bird wing when under aerodynamic loading [20]. The profile of the upper side as well as from the lower side were obtained at specific locations of the wing-span. A laser-line device (Diode Pumped Solid State (DPSS) laser, Nd: YAG (fx2), monochromatic light λ = 523 nm) was positioned outside of the wind tunnel projecting a laser sheet parallel to the main axis of the wind tunnel. The device was mounted on a rail allowing it to be positioned over the whole wing-span. The upper and lower side of the wing were illuminated at the same time by positioning a mirror on the section floor which reflected part of the laser line coming from above. The set-up used for the scanning method is shown in Figure 4.

Two GoPro^®^ Hero 7 cameras @ 125 fps were attached inside the wind tunnel section in the far upper and lower corner and positioned to capture the green line of the laser projected into the wing. Picture distortion due to the 15° view angles of both cameras were corrected afterwards. Recording of the projected laser lines was done after darkening the room. To obtain the coordinates of the upper and lower profiles of the wing, a Python script was written. The code first turned the images into gray-scale since this allows to filter the pixels based on threshold values. Once the threshold is set, the pixels having any color were turned into white color and pixels outside the range were turned into black color. Therefore, a pure black and white image was generated, see Figure 5c. The coordinates of the white pixels were obtained by using the smooth function of a generalized additive model (GAM) [21]. A GAM combines linear functions with flexible-non-linear functions resulting in a smoother line (spline) that depicts the non-linear shape of the wing profiles. Lastly, the cross-section was formed by joining the last coordinates of the trailing edge of both profiles assuming they were in the same position. A schematic positioning of the cameras, the projection of the laser on the upper and lower profiles, and the processed images generated by the Python code are shown in Figure 5.

The coordinates files generated from the laser scan method were imported to the computer aided design (CAD) software Autodesk Fusion 360©. This CAD served as an editing tool for the construction of the 3D blade model. The leading edge of each profile was constructed by drawing an arc following the curvature of the coordinates through the most forward points of both profiles. The closed profiles were located at the same distance from the base as when scanned. The scanned profiles showed the outline of the morphology of the bird wing in cross section and in some cases also interruptions between feathers. Therefore, it was decided to select known database profiles which fit as much as possible to the wing profile scans. Similar to previous studies, the decision was made to represent the full blade by 3 profiles that fluently connect to one another [22]. The three profiles were each derived from a group of three neighboring scans, resulting in one profile for the three scan locations of the base section, similar for the middle section and similar for the tip section. Per section, the chord, camber and thickness of the three scans for that section were compared to known database-profiles, see Figure 6, [23]. The final profiles can be found in Table 2. The fluent connections between the profiles ID 3 and 4 and between profile ID 6 and 7 were created using the loft function of the CAD software. The chord of the profiles near to the base was increased to fit the base hub area adding stability to the final model. Lastly, after scaling, the profiles were lined up, attached to the hub base and printed in a Form 3+, Formlabs^®^ printer.

### 2.4. Flow Tank Measurements

As a first approach of studying the behavior of the bio-inspired blades in a marine environment, they were assembled to a marine turbine model and then tested in a large flow tank. The tests were performed in the flow tank facility at the Laboratory of Coasts and Ports located in the Institute of Engineering, UNAM, Mexico City. The flow tank is 37 m long, 0.8 m wide and 1.2 m depth. The facility is capable to simulate currents up to 0.7 m s^−1^ and a wide range of wave spectra [24]. However, for the present study only the variable flow speed function was used. The flow tank facility can be seen in Figure 7.

The marine turbine model has a three bladed horizontal axis rotor with a diameter of 0.31 m. The size was considered to fit one third of the width of the channel in order to avoid blockage effects from the side walls. The materials utilized were polylactic acid (PLA), acrylic transparent and polyvinyl chloride (PVC) pipe tube materials. The PLA material components as the nacelle, hub, lid, acrylic receivers and couplers were printed in a fused deposition modelling (FDM) printer Ultimaker^®^ Extended 2+. To waterproof the inner part of the turbine model, a blender blade mechanism and a rubber seal were used. One side of the stainless-steel shaft was connected via a coupler to the blender mechanism and in turn, via a second coupler, to the gear motor functioning as a generator (Pololu 37D Metal Gearmotor, 30:1 metal gearbox, 37D × 52L mm, 12 V). After printing, the PLA components were sanded and a layer of pool paint was applied to prevent deformations in the material due to water uptake while performing the tests. The marine turbine model and components can be seen in Figure 8.

After leaking tests, a total of 39 test were performed in 3 different flow velocities. To measure inflow velocity, a high-resolution acoustic Doppler velocimeter (Nortek^®^ Vectrino-II Profiling Velocimeter, 70D × 388L mm, 12–48 VDC) was positioned upstream the turbine model, see Figure 9b. Each measurement set consisted of at least 30 s of recording and in order to produce reliable data every set was measured 3 times. The poles of the gear motor were connected to a protoboard in which a set of resistors were selected. Two multimeters were connected to the protoboard to measure resistance and voltage values. The previous set-up description can be seen in Figure 9. Once the data was obtained from the recordings, the power coefficient and tip speed ratio (TSR) were calculated using Equations (3) and (4), respectively.
(3)$$C_{p}=\frac{P_{out}}{P_{in}}=\frac{Mechanical\;power}{\frac{1}{2}\;\rho \;v^{3}\;A}$$
(4)$$TSR=\frac{\omega \;r}{v}=\frac{2\;\pi \;r\;n}{v}$$
where $C_{p}$ stands for power coefficient, $P_{out}$ and $P_{in}$ for power output and input [W] respectively, V for voltage [V], R for resistance [Ω], ρ for water density [kg m^−3^], v for inflow velocity [m s^−1^], A for the rotor area [m^2^], ω for angular velocity [m s^−1^], r for radius of the rotor and n for number of revolutions.

## 3. Results

### 3.1. Wind Tunnel Results

To determine if the wing of the species Common Guillemot *Uria aalge* had similarities in function with a marine turbine blade, wind tunnel tests were performed and its aerodynamic properties were analyzed. As can be seen in Figure 10, for the left wing we found higher lift-to-drag coefficients at the lowest velocity tested, 4 m s^−1^, which indicates lower pressure distributions along the wing-span resulting in higher lift values at lower velocities compared to higher. Our results for the left wing show a promising feature for further analysis in a marine environment. The graph lift-to-drag coefficient against angle of attack (AOA) for the left wing can be seen in Figure 11.

For the right wing, up to 40% higher values were found for all velocities compared with the left wing. The higher lift-to-drag coefficients were found at 12 m s^−1^, slightly higher than 4 m s^−1^, see Figure 12. As the same laboratory set-up was used for both wings’ measurements, the right wing was turned upside down in order to perform the tests. We corrected the measurements for gravity effects but nevertheless found higher values for the up-side-down right wing which will be evaluated in the Discussion section. For now, we consider our results to be adequate for further analysis towards a marine application, with emphasis on the left wing results.

### 3.2. Bio-Inspired Blade Results

Seven profiles were scanned from the left-wing bird model and 3 different known database profiles were selected to construct a practical bio-inspired blade. First, the bird profiles were positioned at the same distance from the base as when scanned. The profile closer to the base, as well as the one closer to the tip, were duplicated in order to reach the total length of the bird wing. Most of the scanned profiles showed irregularities illustrating their biological origin, see Figure 13b. In order to prevent irregular shapes along the blade that could reduce the hydrodynamic performance, known database profiles were selected to create a continuous blade. The scanned profiles were aligned by joining the leading edge and the angle of attack was set. Based on the aerodynamic results, the higher angle of attack was set at the base and decreased up to the tip spanwise resulting a fixed pitch and twist model. In a similar way, the blade was smoothed by increasing the size of the base profile to fit the hub base and decreased in size along the spanwise to the tip, see Figure 13c. The characteristics of the blade can be seen in Table 2. Lastly, the blade was attached to the hub base and 3D printed, see Figure 13d.

### 3.3. Flow Tank Results

The hydrodynamic performance of a marine turbine model was tested in a large flow tank facility. To represent the hydrodynamic efficiency of the rotor, the power coefficient against tip speed ratio were calculated and plotted. We found efficiencies up to 30% for TSR values of ≈3.3, see Table 3. The higher mechanical power values registered in each set of flow velocities were plotted and compared with theoretical predictions following the Betz limit. We observed up to 53% of the maximum power that can be extracted by the marine turbine model, see Figure 14.

## 4. Discussion

In this study, we have followed a biomimetic pathway to analyze left and right, bird wings of the species Common Guillemot (*Uria aalge*) for application in a marine turbine. We have analyzed aerodynamic properties of both wings by performing wind tunnel laboratory measurements. We found higher lift-to-drag coefficients at the lowest velocity tested for the left wing. Based on our results, we derived a bio-inspired blade model by following a laser scanning method, adapting known database profiles and 3D printing the blades. The bio-inspired blades were attached to a marine turbine and tested in a large flow tank facility. We found efficiencies ($C_{p}$) up to 0.3 which is around 53% of the maximum power that can be expected from a turbine model according to the Betz approach. We consider the sub-maximal performance of our biomimetic MEC to be due to sub-optimal settings of the blade pitch and the blade twist. This is illustrated by the relatively low TSR that was registered and the accompanying low $C_{p}$. Several studies have shown that when varying blade pitch and blade twist towards optimal values, maximum power values are obtained [25,26]. Therefore, we expect that when varying blade pitch and blade twist our biomimetic blades to perform closer to the Betz limit.

Despite careful experimental procedures we faced certain limitations. In the case of the wind tunnel experiment, we measured higher lift-to-drag coefficients for the right wing when compared with the left wing. Although we applied gravity compensation steps, the gravity force in addition to vibrations of the set-up assembly could have caused the increase of the lift-to-drag ratio and the wider range of standard deviations. In previous studies where the same methodology was followed, higher values, up to 3 times higher lift values of the right over the left wing were reported [10,11,12]. Our results show a 1.5-fold increase. In all the studies where we encountered this difference, we could not find a causality for this. In hindsight, gravity might have caused a small re-arrangement of the feathers causing different lift forces as well as high flutter.

The flow tank experiments faced limitations in the range of velocities the facility can reach. The maximum flow speed is up to 0.7 m s^−1^ at a water depth of 0.4 m [24]. At higher depth the maximum velocity goes down proportionally. The rotor diameter was covering one third of the width of the channel cross-section, making sure that we did not suffer from blockage effects. At the depth of 0.6 m, which we used, the velocity range was 0–0.54 m s^−1^. Our net efficiency results are not far from previously reported values [7] with $C_{p}$ of 0.4–0.5. Nevertheless, there is still room for improving our bio-inspired design procedure and/or experimental procedures. We consider further research is necessary for better understanding the hydrodynamic performance of our potentially promising marine turbine model.

## Figures and Tables

**Figure 1 biomimetics-08-00118-f001:**
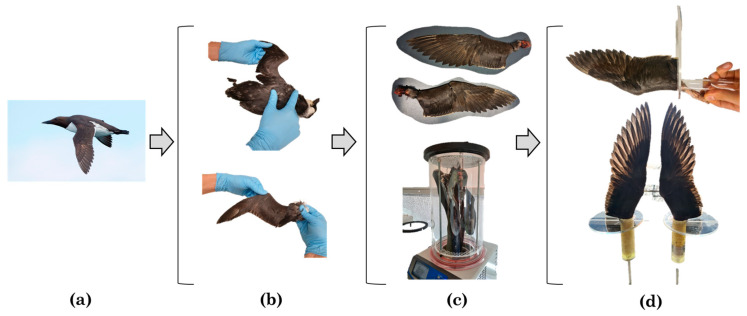
Wing preparation process: (**a**) Selection of the bird Common Guillemot (*Uria aalge*) [14], (**b**) Obtaining the wing, (**c**) mounting and freeze-drying process, (**d**) Fixation of the wings.

**Figure 2 biomimetics-08-00118-f002:**
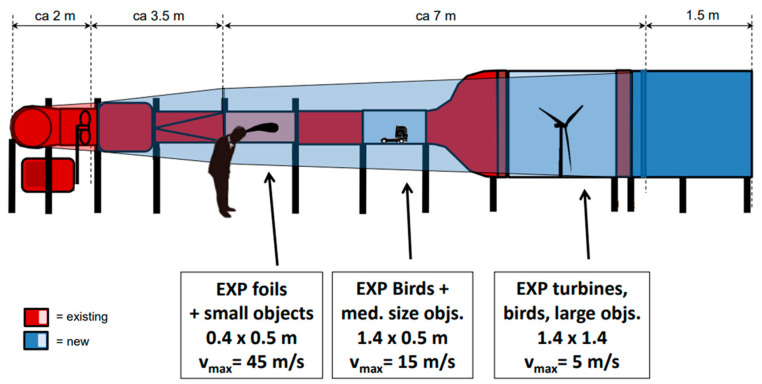
Side view of Wind Tunnel facility. Modified from [16].

**Figure 3 biomimetics-08-00118-f003:**
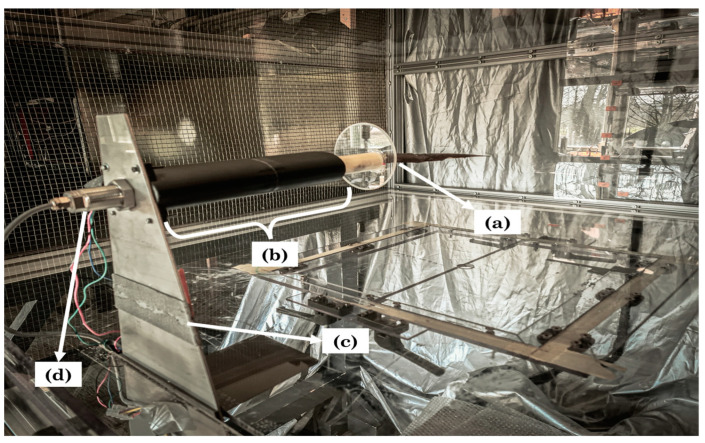
Set-up of wind tunnel measurements: (a) Bird wing fixation, (b) 3D printed streamliner covering the stepper motor and force sensor, (c) Thick metal plate, (d) outward side of the plate directing wires through the slit to control equipment outside the channel.

**Figure 4 biomimetics-08-00118-f004:**
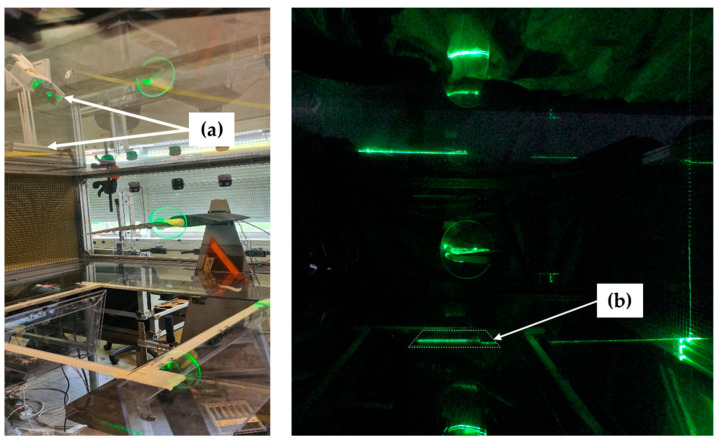
Set-up of the laser scan methodology: (a) Laser-line device and rail, (b) Surface mirror on the section floor.

**Figure 5 biomimetics-08-00118-f005:**
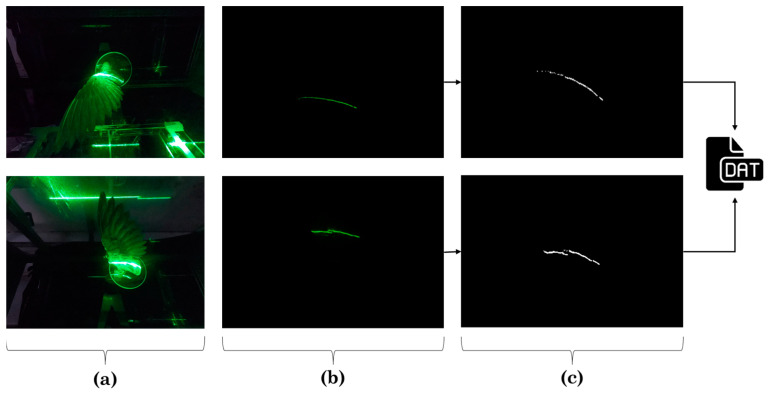
Laser scan methodology: (**a**) Schematic positioning of the cameras, (**b**) Projection of the laser, (**c**) Phyton code black and white images.

**Figure 6 biomimetics-08-00118-f006:**
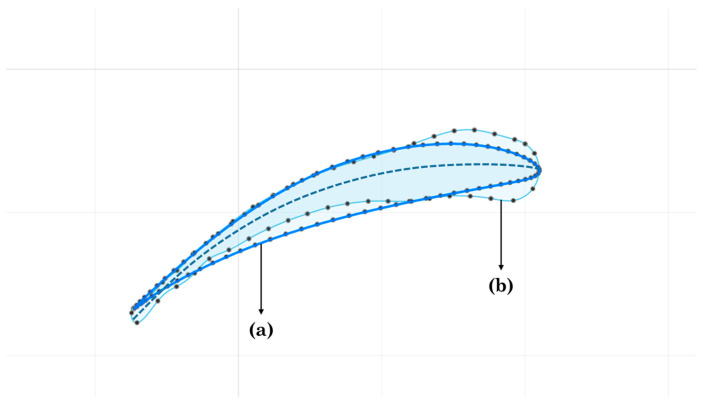
Fitting known database profiles procedure: (**a**) airfoil CH10, (**b**) scanned section of the bird wing at 3 cm from the base. Screenshot taken from Autodesk Fusion 360 © 2020 Autodesk, Inc.

**Figure 7 biomimetics-08-00118-f007:**
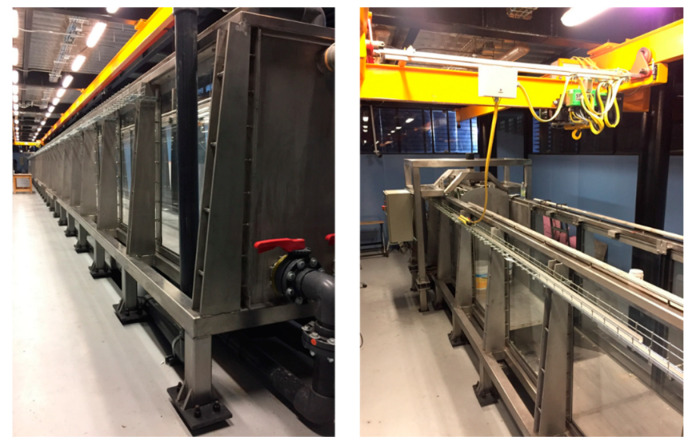
Flow tank facility, Laboratory of Coasts and Ports, UNAM, Mexico City.

**Figure 8 biomimetics-08-00118-f008:**
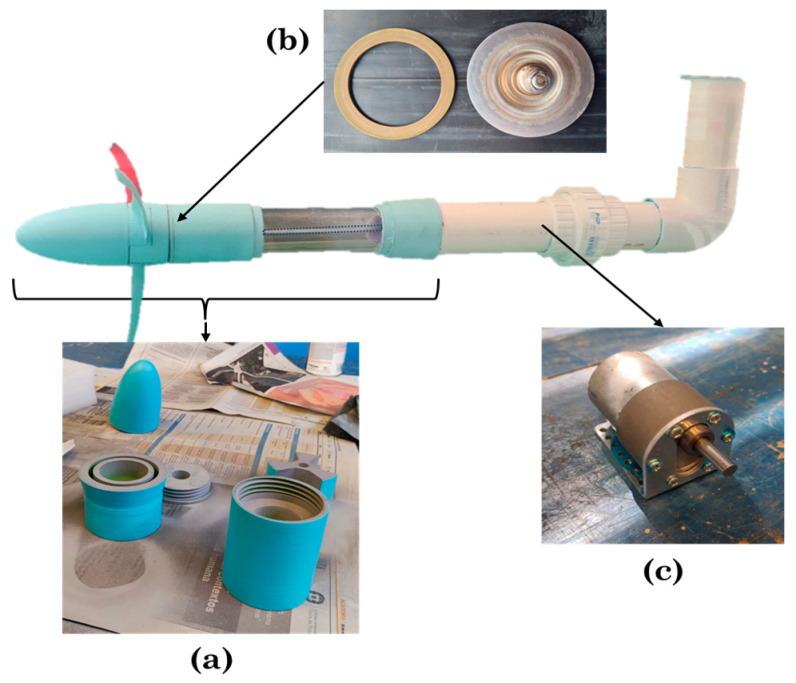
Components of the marine turbine model: (**a**) PLA components, (**b**) Blender blade mechanism and rubber seal, (**c**) Gear motor.

**Figure 9 biomimetics-08-00118-f009:**
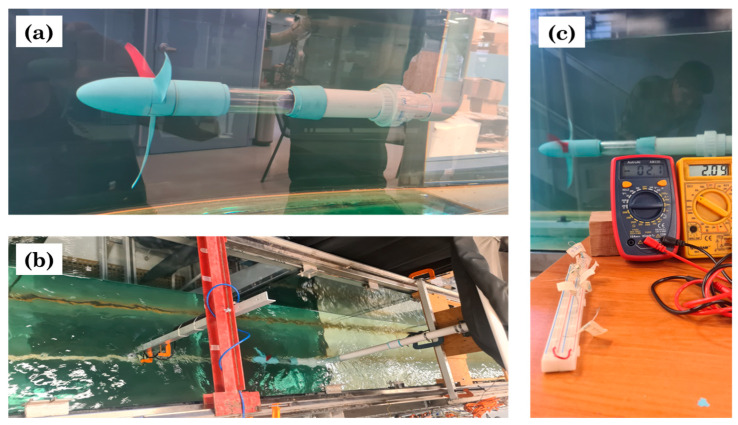
Flow tank tests set-up: (**a**) Leak tests, (**b**) Upstream positioning of measurement device, (**c**) Multimeters and protoboard connected to the gear motor.

**Figure 10 biomimetics-08-00118-f010:**
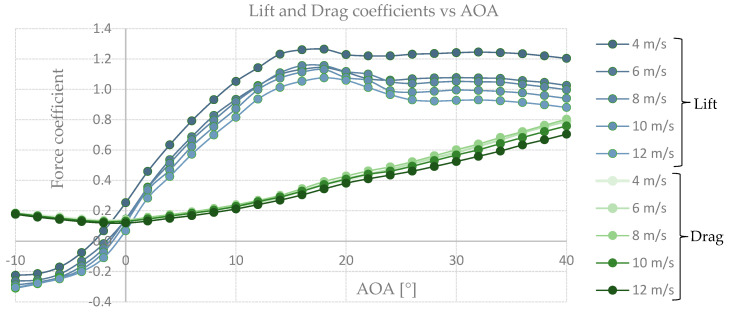
Lift and Drag coefficient against angle of attack.

**Figure 11 biomimetics-08-00118-f011:**
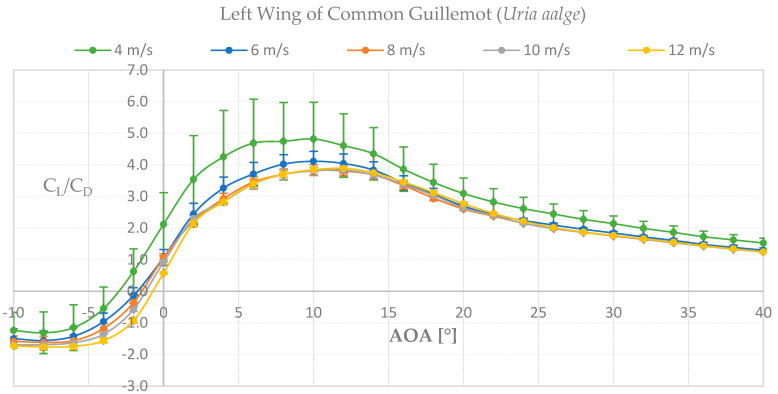
Force measurements of lift−to−drag coefficient against angle of attack for the left wing.

**Figure 12 biomimetics-08-00118-f012:**
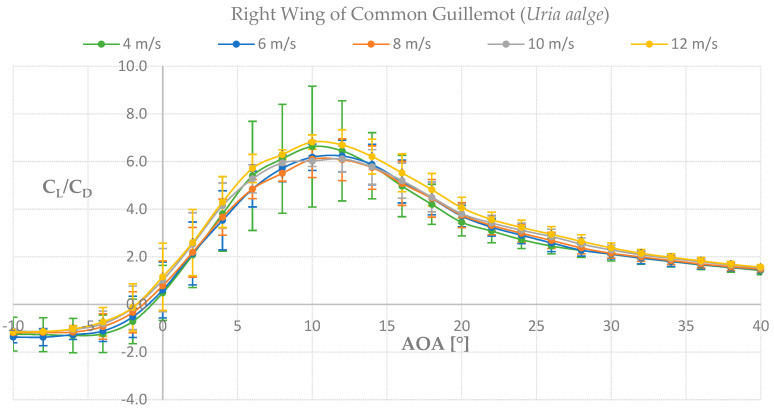
Force measurements of lift−to−drag coefficient against angle of attack for the right wing.

**Figure 13 biomimetics-08-00118-f013:**
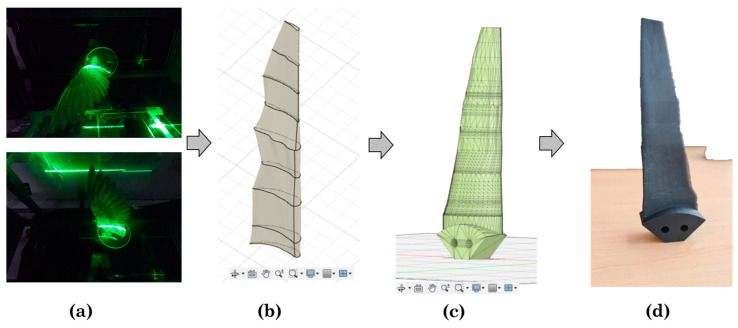
Bio-inspired blade processes: (**a**) Laser scanning method, (**b**) Scanned profiles from the bird wing imported and positioned in Autodesk Fusion 360 ©, (**c**) Line up and adjust of selected known database profiles, (**d**) Scaled 3D printed model.

**Figure 14 biomimetics-08-00118-f014:**
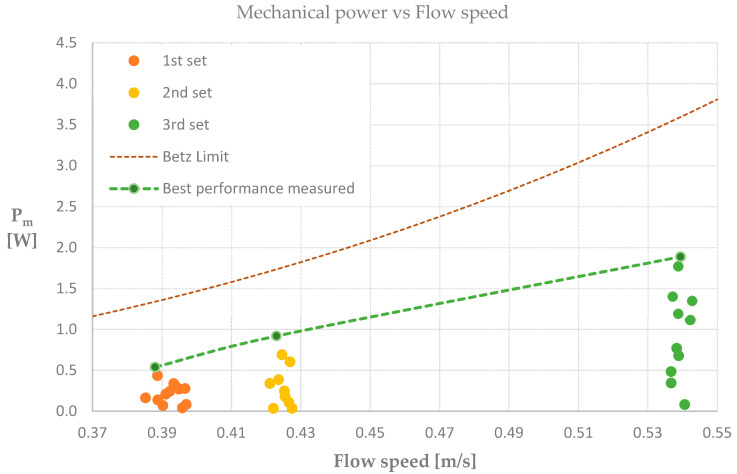
Mechanical power compared to Betz limit theoretical values.

**Table 1 biomimetics-08-00118-t001:** Characteristics of the wings during the freeze-drying method.

	Weight R-Wing [g]	Weight L-Wing [g]
Initial wing mass	21.19	20.65
Wing mass after 12 days	10.68	10.15
Mass Loss [%]	50.4	49.1

**Table 2 biomimetics-08-00118-t002:** Bio-inspired turbine blade characteristics.

Profile ID	Airfoil	Distance from Base [cm]	Twist [°]
1	CH10	0	12
2	CH10	3	12
3	CH10	5.5	10
4	NACA–4412	8.5	8
5	NACA–4412	11.5	6
6	NACA–4412	14.5	4
7	DGA1138	17.5	3
8	DGA1138	20.5	1
9	DGA1138	24.3	0

**Table 3 biomimetics-08-00118-t003:** Power coefficient and tip speed ratio mean values.

Flow Speed [m/s]	$\mathrm{Power}\;\mathrm{Coefficient}\;\left[C_{p}\right]$	Tip Speed Ratio [TSR]
0.37–0.39	0.17 ± 0.06	3.3 ± 0.25
0.42–0.44	0.20 ± 0.08	3.2 ± 0.11
0.51–0.54	0.24 ± 0.05	3.4 ± 0.29
Mean values	0.2	3.3

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
