# Peer review of "Biomimetic Design of Turbine Blades for Ocean Current Power Generation"

_biomimetics, 2023, doi:10.3390/biomimetics8010118_

Round 1

Reviewer 1 Report

Review attached.

Author Response

Dear Reviewer 1,

First, we would like to thank you for your comments, they have been really helpful for improving the presented manuscript. Attached you will find a point-by-point response.

Best regards, 

Enrique Hernández

Reviewer 2 Report

The paper is well written and will be an added value to the community.

I am just concerning about the uncertainity analysis of the whole measurments, so I suggest to add a table for all sensors used in experiments along with all manufactruru data and uncertainity analysis (compound uncertainity).

Some regerenves shoyld be added to enhance the literature concerning ghte work done in this manuscript.

"Lift and Drag of Flapping Membrane Wings at High Angles of Attack", Mohamed Y. Zakaria David W. Allen Craig A. Woolsey  and  Muhammad R. Hajj,  AIAA 2016-3554, Flapping Flight Aerodynamics, Published Online:https://doi.org/10.2514/6.2016-3554

Author Response

Dear Reviewer 2,

First, we would like to thank you for your comments, they have been really helpful for improving the presented manuscript.

We included more information on the sensor sensitivity It is also included a reference to the manufacturers database. 

With respect to comparing to more literature: there are hardly any comparable studies especially not with regard to biomimetic tidal flow turbines. However, new references were added in the Discussion section supporting our statements and a new graph was included in the Result section for the same reason.

Best regards, 

Enrique Hernández

Reviewer 3 Report

The paper focuses on experimental research and design of a bio-inspired turbine blades useful for ocean current power generation. The theme is in accordance to the nowadays trend of extending and optimizing the renewable energies exploitation.

The structure and the content of the work is adequate, however few improvements are still recommended:

- line 9: please check the spelling "crucial roll";

- lines 27-28: please check if there is the expected meaning "...governments and industrial sectors have compromised resources and global agreements to upgrade renewable energy."

- line 35: "Nowadays, wind power and solar PV account for 5% and 2.5%" please update the values and also the reference should be "on-line" type with associated link (eg. IEA reports now 3.6% for PV according to https://www.iea.org/reports/solar-pv);

- line 70: typing issue or undefined acronyms present: "c.q. bio-inspiration";

- lines 217-219: the statement might be unclear "The upper and lower side of the wing were illuminated at the same time because a surface mirror on the section floor reflected part of the laser line coming from above." (a reformulation such that to clarify if reflection is desired or not is suggested);

- lines 229-230: the used image processing library can be mentioned;

- line 234 - 240: there is not clear if an existing tool or own developed mathematical equations (if exists they can be included in the paper) and conversion algorithm/code/program was used to obtain the geometric coordinates of the wing surface;

- lines 246 - 247: in the following statement there is about circle or arcs of  circles "The leading edge of each profile was constructed either using a circle following the curvature"? A criterion of choosing the radius can be specified.

- lines 277 - 278: reformulation for a better clarity is suggested (eg. three bladed horizontal axis rotor with the diameter of ...): "The marine turbine model has a rotor diameter of 0.31 m and a three bladed horizontal axis rotor.";

- line 319-320: check and update to the proper tense " ...the aerodynamic properties analyzed.";

- line 320: specifying also in numerical/procentual values would be more easy comparable with other results from the literature;

- line 324: undefined acronym is used (eg. angle of attack (AOA) - is expected);

- state-of-the art and/or discussion section can be extended such that to  expand the bibliographic list with 10-15 additional references;

- for an improved structure of the paper the current Discussion section can be developed with an exhaustive comparisons and comments related other existing solutions and after to introduced a distinct Conclusions section;

- Can the authors motivate the rationale for choosing experimental testing in the first instance? If they built the numerical (digital) model did the authors intend to explore model optimization by FEM or similar types of analysis prior to future experimental verification?

Author Response

Dear Reviewer 3,

First, we would like to thank you for your comments, they have been really helpful for improving the presented manuscript. Attached you will find a point-by-point response.

Best regards, 

Enrique Hernández

Round 2

Reviewer 1 Report

Overall, I am satisfied that the authors have addressed my comments and recommend the paper is accepted. Ideally an additional control experiment should be included for comparison with the novel turbine design, but reference to similar studies in literature suffices.